# An Amplicon-Based Approach for the Whole-Genome Sequencing of Human Metapneumovirus

**DOI:** 10.3390/v13030499

**Published:** 2021-03-18

**Authors:** Rachel L. Tulloch, Jen Kok, Ian Carter, Dominic E. Dwyer, John-Sebastian Eden

**Affiliations:** 1Centre for Virus Research, Westmead Institute for Medical Research, Westmead, NSW 2145, Australia; rachel.tulloch@sydney.edu.au; 2Marie Bashir Institute for Infectious Diseases and Biosecurity, Sydney Medical School, The University of Sydney, Sydney, NSW 2006, Australia; dominic.dwyer@sydney.edu.au; 3NSW Health Pathology-Institute for Clinical Pathology and Medical Research, Westmead Hospital, Westmead, NSW 2145, Australia; jen.kok@health.nsw.gov.au (J.K.); ian.carter@health.nsw.gov.au (I.C.)

**Keywords:** human metapneumovirus, whole-genome sequencing, genomic epidemiology

## Abstract

Human metapneumovirus (HMPV) is an important cause of upper and lower respiratory tract disease in individuals of all ages. It is estimated that most individuals will be infected by HMPV by the age of five years old. Despite this burden of disease, there remain caveats in our knowledge of global genetic diversity due to a lack of HMPV sequencing, particularly at the whole-genome scale. The purpose of this study was to create a simple and robust approach for HMPV whole-genome sequencing to be used for genomic epidemiological studies. To design our assay, all available HMPV full-length genome sequences were downloaded from the National Center for Biotechnology Information (NCBI) GenBank database and used to design four primer sets to amplify long, overlapping amplicons spanning the viral genome and, importantly, specific to all known HMPV subtypes. These amplicons were then pooled and sequenced on an Illumina iSeq 100 (Illumina, San Diego, CA, USA); however, the approach is suitable to other common sequencing platforms. We demonstrate the utility of this method using a representative subset of clinical samples and examine these sequences using a phylogenetic approach. Here we present an amplicon-based method for the whole-genome sequencing of HMPV from clinical extracts that can be used to better inform genomic studies of HMPV epidemiology and evolution.

## 1. Introduction

Since its discovery in 2001, human metapneumovirus (HMPV) has become increasingly recognised as a major cause of acute respiratory illness (ARI), globally [1]. Serological studies estimate that almost all individuals will be exposed to HMPV by the age of five [2]. Clinically, HMPV is indistinguishable from ARI caused by other respiratory pathogens, including respiratory syncytial virus (RSV) [3]. While most infections are mild and self-limiting, HMPV has increased morbidity and mortality in high-risk populations, including immunosuppressed individuals, young children and the elderly [4]. Globally, HMPV is associated with 3.9–7% of children hospitalised with lower respiratory tract infections (LRTI), with outpatient detection rates ranging from 6.2 to 12%, highlighting its clinical significance as a cause of ARI in this age group [3,5,6,7]. HMPV is also a known cause of ARI in hospitalised adults, with one study showing that the hospitalisation rates of adults >50 years of age were statistically similar to those with influenza infections in the same region [8].

HMPV is a member of the *Pneumoviridae* family, with a negative-sense, single-stranded RNA genome of approximately 13.3 kb in length [9,10]. HMPV is genetically similar to RSV; however, it lacks two nonstructural genes—NS1 and NS2. Phylogenetic analysis of the HMPV fusion (F) and glycoprotein (G) genes has led to the identification and classification of viruses into two major subtypes, HMPV A and HMPV B, which can further be subdivided into A1, A2a, A2b, B1 and B2 sublineages [11]. Epidemiological studies have revealed the cocirculation of HMPV subtypes globally, with the predominant subtype fluctuating throughout the year [12]. Historically, HMPV molecular epidemiological studies have relied on the subgenomic sequencing of partial F or G protein genes to perform genomic and evolutionary studies [13]. Indeed, only 2.3% (n = 226/9795) of available sequences on the National Center for Biotechnology Information (NCBI) GenBank database are near-complete or complete genomes (as of November 2020). Therefore, our understanding of the genomic epidemiology, genetic diversity and evolution of HPMV remains limited.

Whole-genome sequencing (WGS) is a powerful tool for public health infectious disease surveillance; it can also inform the treatment and control of viruses in the population [14,15]. WGS offers increased resolution at multiple epidemiological scales, from investigating global virus traffic networks to elucidating individual transmission events within outbreaks [15,16]. The recent SARS-CoV-2 epidemic has highlighted the utility of amplicon-based WGS methods as a cost-effective, rapid method to sequence the whole-genome approach [17,18,19]. The purpose of this study was to develop a simple and robust amplicon-based method for sequencing the HMPV full-length genome with the aim to inform a better understanding of its molecular epidemiology.

## 2. Materials and Methods

### 2.1. Primer Design

Our approach was based on an existing WGS workflow designed to amplify and sequence the RSV genome using four amplicons between 3,528 and 4,375 nt in length, each with an overlapping region of at least 100 nt [20]. Given the similar genome lengths between HPMV and RSV, we focused on designing HMPV-specific primer sets that would also generate four overlapping amplicons of ~3.5 kb each that span the viral genome. To include historical and current circulating HMPV subtypes, all available full- or near-full-length (>13,000 nt) HMPV genome sequences were obtained from the NCBI GenBank database (downloaded on 10 December 2019), and all sequence analysis was conducted using Geneious Prime 2019.2.3. Sequences were excluded from the initial analysis if they contained >20 continuous ambiguous bases. This result was a final set of 153 full- or near-full-length HMPV sequences, which were then aligned using MAFFT version 7.45 [21]. Phylogenetic analysis using a neighbour-joining approach was performed to show that all known HMPV subtypes were represented in the subset of sequences (Figure 1). 

A sliding window approach was then used to plot sequence identity along the viral genome alignment and identify conserved regions for targeted primer design. These primers were designed to be ~25 bp in length with degeneracies where necessary to capture position variation between HMPV subtypes, as well as to have melting temperatures within 5 °C of each other, and avoiding potential dimer formations [22]. We also tested an existing published primer set designed to amplify the HPMV genome using a similar overlapping amplicon scheme (~3.5 kb each) from a study examining the local virus traffic and genetic diversity of HMPV in Peru [23]. The final set of primer pairs was designed to be run as four separate polymerase chain reactions (PCRs) performed in parallel (Table 1) and was tested on a selection of known viral extracts from HMPV-positive clinical specimens.

### 2.2. Clinical Specimens and HMPV Isolation

Residual total nucleic acid from respiratory specimens submitted to NSW Health Pathology, Institute of Clinical Pathology and Microbiology Research (ICPMR) for diagnostic testing were utilised in this study as per protocols approved by the Westmead Hospital and Institute ethics and governance committees (LNR/17/WMEAD/128 and SSA/17/WMEAD/129, 16 November 2017). Total nucleic acid was extracted from each submitted respiratory specimen using a high-throughput, magnetic-bead-based extraction platform and screened against a panel of known respiratory viruses using an accredited multiplex quantitative RT-PCR (qRT-PCR). Viruses included on the respiratory panel are influenza A, influenza A subtype H3N2, influenza A H1N1 2009 pandemic, influenza B, adenovirus, parainfluenza 1, 2 and 3, RSV, rhinovirus, enterovirus and HPMV. Archived clinical extracts that were reported as positive for HMPV were deidentified before inclusion in this study. No specific subtyping information was available on the archived samples; therefore, we selected ten random HMPV-positive nucleic acid specimens collected over seven years between 2013 and 2020 to attempt to capture historic HMPV genetic diversity in NSW, Australia. To estimate the levels of HMPV, an in-house singleplex qRT-PCR specific for HMPV was performed, with CT values ranging between 19.9 and 34.2 in our study samples.

### 2.3. Reverse Transcription

Complementary DNA (cDNA) synthesis was performed in a 20 µL reaction containing 4 µL of 5× SuperScript IV VILO MasterMix (Invitrogen, Carlsbad, CA, USA), 12 µL of nuclease-free water and 4 µL of the viral RNA template, as per the manufacturer’s instructions. The thermal cycling protocol used was as follows: random priming was performed at 25 °C for 10 min, followed by extension at 50 °C for 20 min and then enzyme denaturation at 85 °C for 5 min before holding at 4 °C. All incubation steps and reaction components were performed to the manufacturer’s specifications on a SimpliAmp thermocycler (Applied Biosystems, Foster City, CA, USA). Viral cDNA was used immediately or stored at −80 °C until required.

### 2.4. HMPV Genome Amplification

The viral cDNA was then split across four separate PCR reactions, each amplifying one part of the HMPV genome (Table 1). Each PCR was performed in a 25 µL reaction containing 12.5 µL of 2× Platinum SuperFi MasterMix (Invitrogen), 1.25 µL of 10 µM forward primer, 1.25 µL of 10 µM reverse primer, 7 µL of nuclease-free water and 3 µL of cDNA template. The reactions were then incubated at 98 °C for 30 s, followed by 44 cycles of denaturation at 98 °C for 10 s, annealing at 60 °C for 20 s and extension at 72 °C for 2:10 min, with a final extension for 5 min at 72 °C, followed by holding at 4 °C. Amplicon size and yield were assessed by gel electrophoresis of 5 µL of PCR reactions using a 1% E-Gel-48 Agarose Gel (Invitrogen) with 500 ng of 1 Kb Plus DNA Ladder. The gels were prestained with ethidium bromide for amplicon visualisation under UV light for approximate quantification and sizing. Amplicon approximate quantity was estimated using the target PCR product band intensity. To ensure even coverage across the HPMV genome, the 4 amplicons of each clinical sample were pooled equally based on target amplicon intensity into a final pooled volume of 40 µL, adjusted with Qiagen EB buffer (Qiagen, Düsseldorf, Germany) if necessary. When nonspecific amplification was present, the band intensity of only the target amplicon was taken into consideration, and pooling was adjusted accordingly. The HMPV genome amplicon pools were purified using AMPure XP (Beckman Coulter, Pasadena, CA, USA) at a bead-to-sample ratio of 1× according to the manufacturer’s protocol. The purified DNA was then quantified using the 1× double-stranded DNA high sensitivity (1× dsDNA HS) Qubit assay (Invitrogen) and measured on the Qubit 4 fluorometer. The pooled amplicons were then volumetrically diluted to 0.2 ng/µL, the required input concentration for library preparation.

### 2.5. Library Preparation and Sequencing

Amplicons were prepared for sequencing using the Nextera XT DNA Library Preparation Kit with the v2 Set B indexing kit (Illumina, Massachusetts, MA, USA), although any compatible indexing set of choice could be used in the reproduction of this approach. The manufacturer’s protocol was followed for genomic DNA tagmentation, library amplification and clean-up, except that all volumes were halved for reagent conservation. The purified DNA libraries were quantified using the 1× dsDNA HS Qubit assay and Qubit 4 fluorometer before normalisation using Qubit DNA concentrations. The final library pool molarity and fragment length distribution were determined using the 4200 TapeStation System with a high sensitivity D5000 tape (Agilent, Santa Clara, CA, USA) before dilution to 0.1 nM with Qiagen EB buffer (Qiagen, Düsseldorf, Germany) for loading and sequencing on an Illumina iSeq 100 platform (Illumina, San Diego, CA, USA) with a v1 300 cycle kit. 

### 2.6. Viral Assembly

Raw sequences were initially quality scored using fastqc [24] following this, the reads were quality trimmed to a QC threshold of phred score 20 using bbduk [25]. The trimmed reads were then de novo assembled using Megahit with default parameters [26]. The trimmed reads were then remapped onto the draft genome using bbmap [25] before the overall mapping alignment quality was assessed using the Geneious Prime 2019.2.3 and majority consensus genome extracted. The final genome was trimmed of terminal primer sequences and annotated using NCBI GenBank reference sequences.

### 2.7. Phylogenetic Analysis

Phylogenetic analysis was performed by aligning all sequences generated in this study against a reference set obtained from NCBI GenBank using MAFFT and PhyML [27], with node support estimated by 100 bootstrap replicates. Sequences obtained in this study were published to NCBI GenBank with the following accession IDs: MW221986-MW221995.

## 3. Results and Discussion

### 3.1. Designing Primers to Amplify the HMPV Genome

The aim of this study was to develop a simple and robust amplicon-based approach for amplifying and sequencing the HMPV genome. To do this, we adapted a previous approach used for RSV [20] to design four primer sets generating ~3.5 kb amplicons that overlap and span the viral genome. Our primers were based on all available HMPV genomes from the NCBI GenBank database and targeted conserved regions at suitably spaced positions in the genome (Table 1 and Figure 2).

The final primer sets were located in the terminal regions of the genome, as well as in the fusion and large protein genes, and avoided divergent regions of the genome such as in the viral glycoprotein (G protein) (Figure 2). Other amplicon-based methods for viral genome sequencing often employ shorter amplicon lengths (1000 bp or less) to improve performance for low-viral-load or low-quality samples, such as with the ARTIC protocol for SARS-CoV-2 genome sequencing [19], or even for enteric virus including human norovirus [28]. Here we chose targeted amplicons to be between 3000 and 4000 bp based on previous performance against RSV [20], where genomes would reliably amplify from 80 to 90% of clinical samples. Furthermore, there is greater diversity present in HMPV compared to SARS-CoV-2 such that there are less suitable target positions across the genome to readily amplify all subtypes. Indeed, to capture this diversity a number of degenerate nucleotides were included in our primers, and based on our current understanding of HMPV diversity, it would be expected our primers cover the vast majority of variants present in circulation (Figure 3).

### 3.2. HMPV Genome RT-PCR Performance

To examine the performance of our newly designed primers, we tested them against a set of HMPV-positive extracts from clinical respiratory specimens. Since the subtype and sublineage classifications from our samples were unknown, we instead obtained samples across a wide time period (2013 to 2020) to attempt to capture a breadth of diversity. Initial end-point PCRs showed good levels of amplification across the four targets (data not shown); however, we then attempted further optimisation of the assay using a temperature gradient (59–61.5 °C) to establish the optimal annealing temperature was 60 °C to ensure efficient target amplification and minimise nonspecific amplification (Figure 4A). We also compared our primers to those previously published from the Peru WGS study (Figure 4B) and showed improved performance, particularly for the specific amplification of the targeted HMPV region; however, this may be partly due to our initial optimisation of annealing and cycling conditions favouring our newly designed set. An additional RT-PCR was performed to ensure the assay was specific for HMPV and did not amplify the closely related virus, RSV (Appendix A).

### 3.3. Genome Sequencing, Assembly and Analysis

Following the successful amplification of all ten HMPV samples, the four amplicons from each were pooled, purified and sequenced using the Nextera XT library prep kit and an Illumina iSeq 100. In this study, we sequenced the 10 clinical HMPV samples along with libraries from other projects. However, we targeted 100,000 paired reads per HMPV library to achieve an expected genome coverage depth between 800 and 1000×, which is sufficient for calling a consensus genome. Given an Illumina iSeq 100 run would yield a total of 5,000,000 paired reads, it would be possible to reliably multiplex up to 48 HMPV genomes per run. In this study, the samples included had a mean read depth of 1125× across all samples Appendix A), and the coverage was found to be even across the genome, except where amplicon pooling was not equal. We then used a de novo assembly approach to generate the final consensus genomes for the ten HMPV samples. Reference mapping would also be an appropriate method for genome assembly; however, similar to RSV, there are notable structural variants (insertions) in the HMPV G protein [20] that may be misassembled when using an inappropriate reference strain for mapping such as a historical prototype. Therefore, a de novo approach would be recommended for both RSV and HMPV WGS, and users should examine coverage profiles for depth variability as an indication of structurally misassembled genomes. It is also important to note that the final genome sequences generated using our approach will be in-complete in the terminal regions and missing an expected 43 and 28 base pairs in the 5′ and 3′ ends, respectively. Ideally, our assay would have had primers designed from the very terminal regions; however, the limited sequence availability meant that such primers using existing GenBank sequences may not capture all circulating diversity. Importantly, the slightly inward placement of primers was not found to have any impact on the phylogeny of the virus, and the topology of phylogenetic trees using complete and “near-complete” genomes (i.e., just using the region our assay amplifies) were found to be identical (data not shown). Furthermore, it is also common practice in phylogenetic analysis to ensure sequences are trimmed to coding regions only, which our assay captures. This genome sequencing approach is also useful in investigating minor variants present in individual patient samples. Previous studies of other respiratory viruses have shown the merit of using amplicon-based methods to identify mixed infections containing multiple viral subtypes [17,20]. However, here in our study samples, no variants were observed at a frequency above 1%. 

We then analysed our ten clinical HMPV genome sequences alongside four reference sequences and a selected subset of published hMPV sequences using a phylogenetic approach to determine their subtype. Of the samples presented in this study, four were identified as A2b strains, two as B1 and four B2 (Figure 5). There are limited data on the molecular epidemiology of HMPV in Australia and nothing published previously for the state of NSW, where these samples were collected. The finding of no A1 or A2a strains may be due to the undersampling in this current study; however, one study from Queensland, Australia, showed declining levels of A1 over the period 2001–2004 [29], and since 2006, these subtypes have been rarely identified with A2b and B strains most commonly identified [30]. Despite this, based on our alignments, we would expect the primers and amplification to capture all subtypes, including A1 and A2, as these viruses were represented in our genome alignments (Figure 1 and Figure 3), and this approach would be useful for ongoing genomic studies here in Australia and globally.

## 4. Conclusions

Using publicly available genome sequences representing the full known diversity of HMPV, we designed a simple and reliable assay for amplifying and sequencing HMPV genomes from clinical samples. Ten HMPV genomes were generated from residual-diagnostic specimens using this approach to demonstrate multiple subtypes circulating in NSW, Australia, since 2013. This work highlights the utility of amplicon-based sequencing for genomic epidemiological studies of respiratory viruses to inform public health investigations and understand the patterns of evolution and spread.

## Figures and Tables

**Figure 1 viruses-13-00499-f001:**
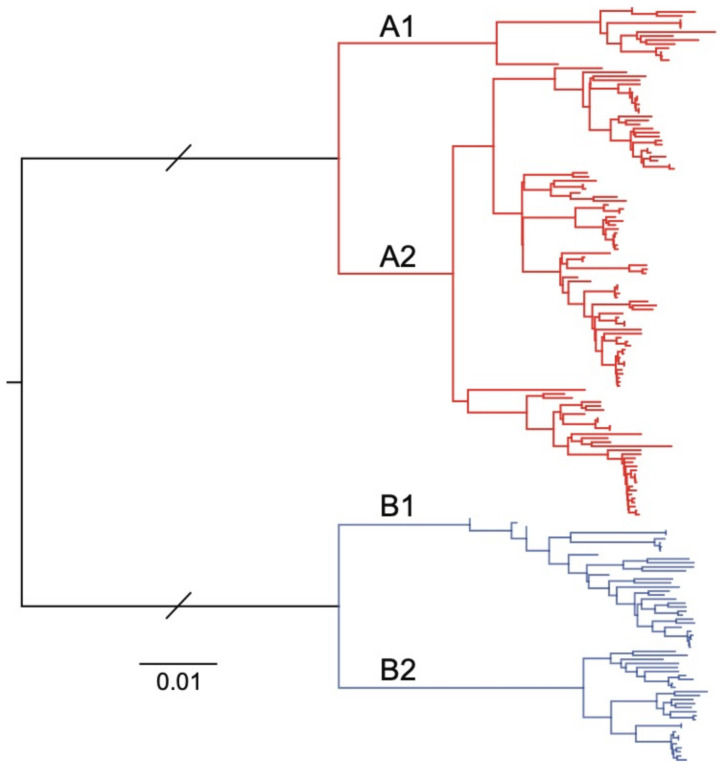
Neighbour-joining phylogenetic tree of 153 human metapneumovirus (HMPV) full-length or near-full-length genome sequences sourced from the National Center for Biotechnology Information (NCBI) GenBank database. The two major phylogenetic subtypes of HMPV are shown with different coloured branches, with known sublineages further differentiated by labelled branches (A1, A2, B1 and B2). The scale bar is proportional to the number of substitutions per site. The branch lengths between the A and B subtypes have been shortened for clarity.

**Figure 2 viruses-13-00499-f002:**
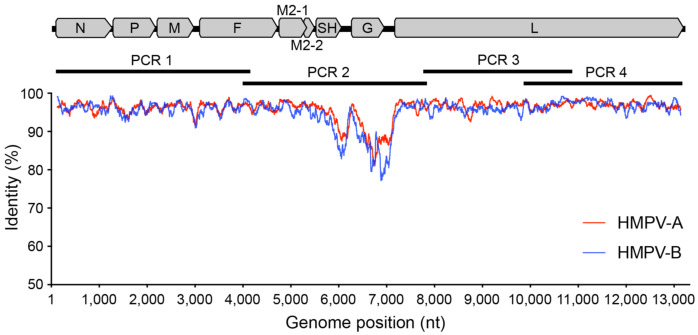
Genetic variability across the HPMV genome. A graphic representation showing the percentage identity at each nucleotide position along the genome with each HMPV subtype plotted separately—A (red) and B (blue). Included also is an HMPV genome shown to scale with annotated genes to highlight primer and amplicon positions.

**Figure 3 viruses-13-00499-f003:**
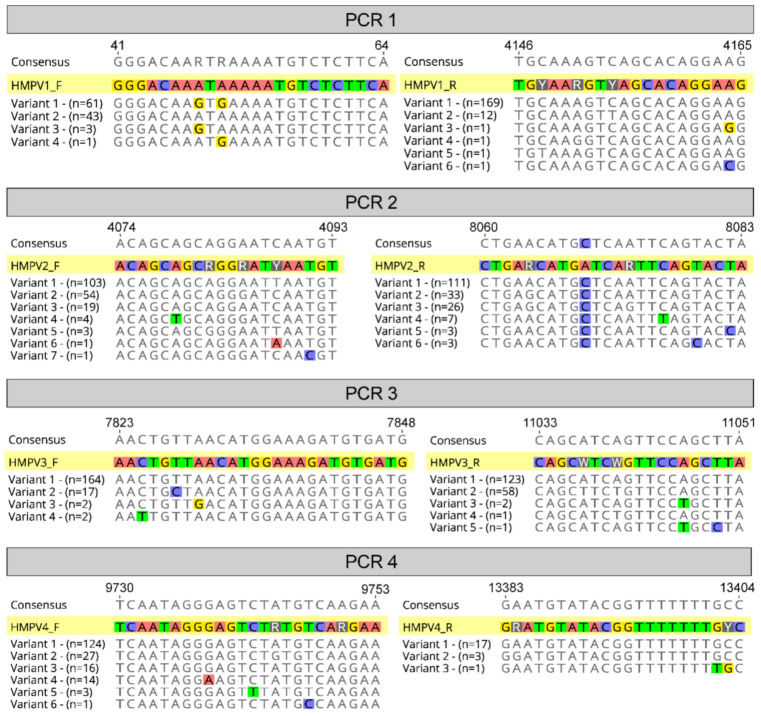
Primer alignments. A graphical representation of the four chosen primer pairs aligned against known variants (sequences in the alignment) at each binding site relative to strain A1 (NCBI GenBank Accession: KU821121). Variants are different possible sequences in the alignment, and “n” refers to the number of these sequences that were observed at the primer binding sites.

**Figure 4 viruses-13-00499-f004:**
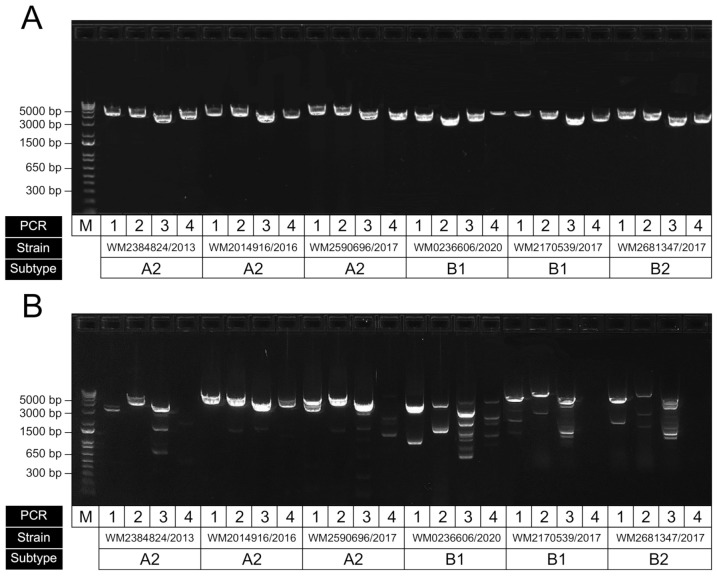
Representative gel electrophoresis result showing RT-PCR performance of amplicons for HMPV genome sequencing. (**A**) Results for our optimised assay using our newly designed primers; (**B**) same samples amplified using previously published primers using the same conditions (Pollett et al., 2018). For each panel, the four amplicons for each sample have been run in sequential order, with strain ID corresponding to the original sample accession and collected year. The samples shown here were of the subtypes A2, B1 and B2, as indicated.

**Figure 5 viruses-13-00499-f005:**
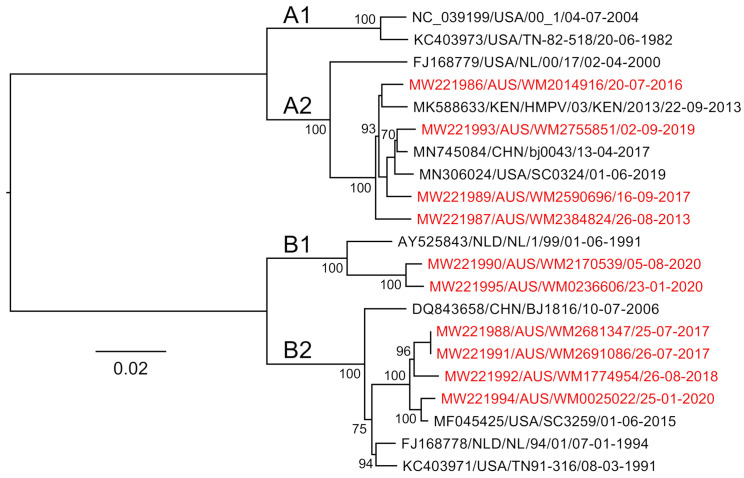
A maximum-likelihood tree constructed using near-full-length human metapneumovirus sequences generated in this study (those colored red). Node supports are indicated, and branch lengths are scaled according to sequence divergence (substitutions per site).

**Table 1 viruses-13-00499-t001:** Primer pairs used in this study to complete the near-whole-genome sequencing of human metapneumovirus.

Assay	Primer Name	Sequence (5’-3’)	Position (nt) *	PCR Amplicon Size (bp)
PCR1	HMPV1_F	GGGACAAATAAAAATGTCTCTTCA	41	4125
	HMPV1_R	CTTCCTGTGCTRACYTTRCA	4165	
PCR2	HMPV2_F	ACAGCAGCRGGRATYAATGT	4074	4010
	HMPV2_R	TAGTACTGAAYTGAGCATGYTCAG	8083	
PCR3	HMPV3_F	AACTGTTAACATGGAAAGATGTGATG	7823	3229
	HMPV3_R	TAAGCTGGAACWGAWGCTG	11,051	
PCR4	HMPV4_F	TCAATAGGGAGTCTRTGTCARGAA	9730	3675
	HMPV4_R	GRCAAAAAAACCGTATACATYC	13,404	

* Nucleotide positions are relative to strain A1 (NCBI GenBank Accession KU821121).

## Data Availability

The sequences generated in this study have been deposited to the NCBI GenBank database under the accession numbers: MW221986-MW221995.

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
