# Peer review of "An Amplicon-Based Approach for the Whole-Genome Sequencing of Human Metapneumovirus"

_viruses, 2021, doi:10.3390/v13030499_

Round 1

Reviewer 1 Report

This manuscript by Rachel L. Tulloch  et al. describe very interesting data focusing on a novel approach for WGS of HMPV.

This article is important for the virologist researchers, especially the one whom work on respiratory viruses, but deserve numerous modification to deserve publication.

Methods : 

give details about the concentration of hMPV in clinical extract?

What are the primers used in reverse transcription ? What are their concentration ?

For all kits/assays, ensure to give manufacturer's name and location.

Why have the authors decided to analyse clinical strains only and not reference strains ? This have to be tested.

Have the authors ensure themselves of the absence of non-specific amplification? If not, they have to test their approach on RSV-positive samples for example.

Why have the authors used the setB for Illumina Indexes?

The figure 4 is very impressive but the figure 4B have to be replaced by the amplification profile of the samples with the primers and protocol described by Pollett et al. as it is obvious that not optimized protocol will give non-optimal results.

What is the impact on phylogeny for the non-covered regions?

Have the authors studied minor frequency variants? What is the mean coverage/depth of the viral strains ? Could the authors give us a coverage plot (at least in supplementary data)?

All numbers below twelve have to be written in full letters.

Author Response

Reviewer One

  1. Give details about the concentration of hMPV in clinical extract?

We have quantified the levels of HMPV using an in-house qPCR at the Institute of Clinical Pathology & Microbiology Research (ICPMR), from where we sourced the samples. We have amended the manuscript to provide the CT values as an indication of the virus concentrations (see lines 113-114).

  1. What are the primers used in reverse transcription? What are their concentration?

Invitrogen SuperScript VILO IV Master Mix is a commercial reverse transcription master mix and the amounts of primers and components are proprietary, so we cannot provide this information. However, it is widely available, and we followed the manufacturers recommended protocol. The manuscript has been amended to reflect this (see lines 125-126).

  1. For all kits/assays, ensure to give manufacturer's name and location.

This information has been added.

  1. Why have the authors decided to analyse clinical strains only and not reference strains? This have to be tested.

Unfortunately, we do not have access to reference strains. Ideally, yes, we would have a complete panel of prototypes covering known virus diversity, however this was not possible. The virus is not commonly cultured or available commercially (at least in the time we have been given to address these comments). Instead, we chose to use temporally diverse clinical samples, which is more relevant to how this assay will hopefully be used in the future. Furthermore, we had a 100% success rate in amplifying these clinical samples that were from all known subtypes except A1. Care has been was taken with our primer design though, to include all available NCBI sequences including the prototype strain: NC_039199, which is an A1 subtype, and we expect our assay to work equally well for this too. We have previously addressed this in our main text (see lines 263-266).

  1. Have the authors ensure themselves of the absence of non-specific amplification? If not, they have to test their approach on RSV-positive samples for example.

We have addressed this by comparing the amplification of 3 hMPV- and 3 RSV-positive samples using both the HMPV and RSV genome amplification protocols. The primers are specific for each virus. The results have been shown in Supplementary Figure 1 and referenced at lines 219-221.

  1. Why have the authors used the setB for Illumina Indexes?

Set B indexes were utilised to avoid cross contamination between existing indexes used in our laboratory. The manuscript has been edited to reflect the ability of any available indexing set to be used interchangeably to reproduce the study (see line 154).

  1. The figure 4 is very impressive but the figure 4B have to be replaced by the amplification profile of the samples with the primers and protocoldescribed by Pollett et al. as it is obvious that not optimized protocol will give non-optimal results.

The purpose of our study was to create a fully repeatable amplicon based whole genome sequencing protocol for HMPV. The study by Pollett et al. only provided the primer sequences and the names of the two different kits that were used. We therefore do not know any of the reaction components or thermocycling conditions they used and their work is not repeatable. Our intention was not to re-develop and optimise their published assay but to take our own established RSV WGS workflow and update it using HMPV specific primers. We found their primers did not perform well for this purpose so moved forward with new designs.

  1. What is the impact on phylogeny for the non-covered regions?

Our approach amplifies and sequences 99% of the virus genome. No observable differences were found in including the terminal regions in the phylogenetic analysis. This was addressed in our results (see lines 245-249).

  1. Have the authors studied minor frequency variants? What is the mean coverage/depth of the viral strains ? Could the authors give us a coverage plot (at least in supplementary data)?

We did utilise the data to investigate minor frequency variants however none were present in our samples (see lines 3249-253). Following your other suggestion, we have also included coverage plots for all 10 samples included in this study (see Supplementary Figure 2), and included a summary of average coverage depth in the main text (see lines 236-237).

  1. All numbers below twelve have to be written in full letters.

Reviewed all numbers in manuscript and corrected.

Reviewer 2 Report

This study by Tulloch RL and colleagues describes the design of primer sets to amplify and sequence the genome of clinical isolates of human metapneumovirus (HMPV) of different subtypes. The paper is well written and easy to follow and will be useful to sequence HMPV clinical isolates. One issue is that the primers actually do not cover the whole genome as the terminal regions of the genome are missing. It would be of interest to design additional primer pairs that include these terminal sequences. This would allow to detect the potential presence of defective interfering viral genomes in clinical isolates. It was also not clear to me if these primer pairs allow to amplify serotype A1 strains. The authors have to demonstrate that their primer pairs can also amplify the genome of serotype A1 strains. In addition, the authors did not show how their deep sequencing data covered the whole genome. Did they find regions of the genome that were cover more or less efficiently than others?

Additional comments:

1-Figure 3: Why did the authors choose a B2 strain as reference? What are the variants indicated in this figure and what “n” means?

2- Figure 4: why no A1 subtype is shown here? Did the authors confirm that their primers pairs can amplify the genome of HMPV A1 strains ?

Author Response

Reviewer Two

  1. Figure 3: Why did the authors choose a B2 strain as reference? What are the variants indicated in this figure and what “n” means?

There was no particular reason for using a B2 strain as the reference. We have amended things to reference an A1 strain (NCBI KU821121) although not the prototype because it seems to lack part of the terminal 3’ end sequence (which are often missing from other strains). Variants are the different possible sequences in the alignment and “n” refers to the number of these sequences that were observed at the primer binding sites. We have revised the legend of Figure 3 to explain this better.

  1. Figure 4: why no A1 subtype is shown here? Did the authors confirm that their primers pairs can amplify the genome of HMPV A1 strains?

HMPV A1 subtypes have rarely been identified in Australia since 2006, and no A1 subtype samples were available to be included in this study. As we have mentioned, our primer design captured all available NCBI sequences and all known subtypes including for example, the prototype strain: NC_039199, which is an A1 subtype, and we expect our assay to work equally well for this subtype as the others that were present in the study. We have previously addressed this in our main text (see lines 263-266).

Round 2

Reviewer 1 Report

After this extensive revision of their writing, the authors allowed the manuscript to be suitable for publication.

Author Response

After this extensive revision of their writing, the authors allowed the manuscript to be suitable for publication.

We thank the reviewer for their time and effort.

Reviewer 2 Report

The authors did not fully reply to my comments. As I previously indicated, it would be interesting to design additional primer pairs that include the HMPV terminal sequences and at least try to amplify the whole genome from clinical samples. Did the authors try this option? This could potentially allow to detect the potential presence of defective interfering viral genomes in clinical samples. In addition, did the authors find regions of the genome that were cover more or less efficiently than others? While they indicated the overall coverage it would be interesting for the reader to know how this coverage varies along the genome.

Author Response

  1. The authors did not fully reply to my comments. As I previously indicated, it would be interesting to design additional primer pairs that include the HMPV terminal sequences and at least try to amplify the whole genome from clinical samples. Did the authors try this option? This could potentially allow to detect the potential presence of defective interfering viral genomes in clinical samples.

Editor: if this is not technically possible in a reasonable timeframe,

this could be discussed and presented as one of the remaining questions

and/or limitations of the method that would require further investigation.

We did consider this in our primer design however there were a significant amount of HMPV genomes on Genbank missing the 5’ and 3’ ends, therefore we felt that the primers designed from these regions would not only use a limited amount of data and risk not capturing all circulating diversity.

We have added a comment on this as a limitation in the main text (see lines 287 - 289)

  1. In addition, did the authors find regions of the genome that were cover more or less efficiently than others? While they indicated the overall coverage it would be interesting for the reader to know how this coverage varies along the genome.

Editor: I believe the authors should have this information available,

which could be incorporated to their manuscript

As shown in Supp Fig 2, the coverage across the genomes was very even and we did not see any regions where the coverage differed except for where the pooling of amplicons was not equal or where assembly artifacts were present such as in the G protein duplications.

We have commented on this in the main text (see lines 277 – 278 & 283 – 285).